# Predicting Satiety from the Analysis of Human Saliva Using Mid-Infrared Spectroscopy Combined with Chemometrics

**DOI:** 10.3390/foods11050711

**Published:** 2022-02-28

**Authors:** Dongdong Ni, Heather E. Smyth, Michael J. Gidley, Daniel Cozzolino

**Affiliations:** Centre for Nutrition and Food Sciences, Queensland Alliance for Agriculture and Food Innovation, The University of Queensland, St Lucia, QLD 4072, Australia; d.ni@uq.edu.au (D.N.); h.smyth@uq.edu.au (H.E.S.); m.gidley@uq.edu.au (M.J.G.)

**Keywords:** saliva, spectroscopy, satiety, satiation, chemometrics

## Abstract

The aim of this study was to evaluate the ability of mid-infrared (MIR) spectroscopy combined with chemometrics to analyze unstimulated saliva as a method to predict satiety in healthy participants. This study also evaluated features in saliva that were related to individual perceptions of human–food interactions. The coefficient of determination (R^2^) and standard error in cross validation (SECV) for the prediction of satiety in all saliva samples were 0.62 and 225.7 satiety area under the curve (AUC), respectively. A correlation between saliva and satiety was found, however, the quantitative prediction of satiety using unstimulated saliva was not robust. Differences in the MIR spectra of saliva between low and high satiety groups, were observed in the following frequency ratios: 1542/2060 cm^−1^ (total protein), 1637/3097 cm^−1^ (α-amino acids), and 1637/616 (chlorides) cm^−1^. In addition, good to excellent models were obtained for the prediction of satiety groups defined as low or high satiety participants (R^2^ 0.92 and SECV 0.10), demonstrating that this method could be used to identify low or high satiety perception types and to select participants for appetite studies. Although quantitative PLS calibration models were not achieved, a qualitative model for the prediction of low and high satiety perception types was obtained using PLS-DA. Furthermore, this study showed that it might be possible to evaluate human/food interactions using MIR spectroscopy as a rapid and cost-effective tool.

## 1. Introduction

A wide range of instrumental and spectroscopic methods such as nuclear magnetic resonance (NMR) [1,2], mass spectrometry (MS) [3], and vibrational spectroscopy (e.g., mid-infrared, Raman) [4,5], have been utilized to profile and analyze the biochemical and chemical composition of saliva [6,7]. Vibrational spectroscopy, and in particular infrared (IR) spectroscopy, has attracted the attention of researchers due to its high sensitivity, high speed, and low cost. The utilization of vibrational spectroscopy has been reported in many areas, including medicine, chemistry, forensic, and food sciences, for the measurement of composition and functional properties [8,9,10]. During the last 20 years, advances in instrumentation and computing have allowed for the evolution of diagnostic methodologies based in vibrational spectroscopy including near infrared (NIR), mid-infrared (MIR), and Raman spectroscopy of saliva and other biofluids [11,12,13,14]. 

It is well known that saliva relates to oral physiology and plays an important role in both oral processing and sensory perception of food [15,16,17,18]. Several studies have reported that salivary proteins are not only correlated with the fungiform papillae density of tongues [19], but they also associated with food aroma compounds released during food intake. Salivary proteins also have a role in the mediation of taste components in sensory perception [17,18]. These factors have strong effects on oral food processing, influencing food particle size, food-saliva interactions, and how nutrients are released during the process of food intake [20]. Moreover, recent studies were focused on the effects of these factors (e.g., oral food processing, food particle size, and food sensory perceptions) on food intake, satiation, and satiety [15,21,22]. A recent study [15] evaluated the relationships between oral sensory exposure and hormones involved in longer oral processing and how these relationships might have influenced satiation and food intake in humans [15,21,22]. It is well known that saliva contains information not only about the composition of the food, but also on the biological and physical properties where recently studies [23,24] reported saliva might be associated with body composition and energy expenditure, which were highly associated with satiety. 

Satiation and satiety are defined as perceived fullness feelings during and after a meal and they are considered the main driving forces responsible for the control of eating behaviors in humans [25,26,27]. The roles of satiation and satiety are to modulate daily meal portion sizes and frequency. Satiety has been proposed to be a key factor in controlling food intake with impacts on an individual’s ability to manage their nutrition and body condition (e.g., weight) [25,26,27]. Moreover, satiation and satiety are not only related to food and energy intake, but they also influence psychological status (e.g., emotion and mood) [28,29]. For example, uncontrolled hunger and psychological phenomena, such as feelings of deprivation and cravings, are major difficulties and main reasons for the ultimate failure of keeping a healthy diet [28,29]. Low satiety human phenotypes [30,31] were also reported in the literature and defined as individuals who can recognize appetite sensations, with both low appetite and low changes in appetite sensation. This research suggested that low satiety individuals have lower self-reported satiety than the high satiety individuals. Drapeau and colleagues [30] reported that low satiety phenotype individuals had a lower blood cortisol response during meal intake. However, only limited physiological differences were reported for this phenotype, as it was challenging to identify this phenotype before the experiment. 

The aim of this study was to assess the ability of mid-infrared spectroscopy to predict satiety (and satiation) using the saliva collected from healthy participants. This approach will open the possibility of utilizing rapid and low-cost methods, such as IR spectroscopy, to evaluate human/food interactions (satiation and satiety) as well as to explore the technology as a screening method in other food and/or physiological studies.

## 2. Materials and Methods

### 2.1. Participants and Saliva Collection

A total of 52 healthy participants (31 female and male 21, with an average of 38.1 and standard deviation of ±13.8 years) were recruited through an open advertisement posted around Brisbane city (Brisbane, QLD, Australia). The selection criteria for participants were based on the following parameters: aged between 18 and 70 years, lack of oral cavities or dental diseases, no diabetes, not being pregnant or lactating at the time of the experiments, and not being diagnosed with psychological diseases, such as depression. The ethics for the research was approved by the Sub-Committee for Human Research Ethics of the University of Queensland (approval number: 2019002688). 

Unstimulated saliva, defined as the saliva that continuously bathes the oral cavity without chewing and taste stimulation [32,33], was collected from each participant prior to the satiation and satiety sensory experiment three times during three consecutive days. On the experimental day, no food or drink (except water) were allowed after breakfast (the recommended consumption time for breakfast was 7:00 a.m. for all the participants). Participants were asked to arrive at the sensory lab around 10:00 a.m. after 3 h fasting. Details about the protocols and methods used to collect the unstimulated saliva were reported in a previous paper [34]. In short, a saliva collection suite was provided to each participant, which included a cup of 10 mL mouth rinsing water, a saliva collection tube stored in a beaker with filled with ice, and a spittoon. Participants were asked to rinse their mouths properly with the provided water, spit the water into the spittoon, and avoid swallowing the saliva for 2 min where they could expectorate saliva into the collection tube every 30 s. The expectorated saliva samples were sealed, double wrapped with plastic bags, and transferred into the −80 °C freezer. In total 156 saliva samples were collected (52 participants × 3 days).

### 2.2. Satiation and Satiety Measurements

Three types of plant-based foods were utilized during the satiation and satiety sensory experiments: an apple (Royal Gala variety); a banana (Cavendish variety); and an avocado (Hass variety). The selection of plant-based foods (e.g., vegetables, fruits) used in this study was related to the fact that plant foods take up over two thirds of the everyday diet recommendations worldwide. The key features of these plant-based foods included high dietary fiber and cell wall structuring that tends to induce both strong satiation and long-lasting satiety. The experiments were conducted on three different days (one food type each day). The three plant-based foods were selected as they each provide a different source of energy, such as soluble sugars for the apple, starch for the banana, and lipids for the avocado). These differences might have induced different satiation and satiety in the participants. The experimental design followed in this study not only balanced the variations in nutrients and energy sources, but also allowed for the comparison of different food types. Each food was cut into approximately 3 mm slices and served in a covered plastic container (100 g portions in each container). Food was served ad libitum during 20 min for each participant. Participants could stop eating when they felt comfortably full. The fullness ratings were determined as described in a previously published paper [35]. Participants’ fullness during the meal and for 150 min after the meal were self-evaluated using a 20 cm labelled magnitude scale (LMS) [36]. Fullness ratings at the time points were measured by the distance (cm) between the greatest imaginable hunger point and the participant’s marked point on the scale. The total area under the curve (AUC) of the perceived fullness over time was defined as satiation (AUC during the meal) and satiety (AUC after the meal) [27]. The participants were separated into low or high satiety groups according to the satiety values (AUC). The low satiety perceiver group were defined as those having an AUC lower than 1100 AUC, whereas the high satiety perceiver group contained the participants with an AUC that was higher than 1100. 

### 2.3. Mid-Infrared Spectrum Collection for Saliva

The spectrum of the saliva samples was collected using an MIR spectrometer ALPHA II (Bruker Optics, Ettilgen, Germany) (4000 cm^−1^ to 400 cm^−1^ region). The spectrometer was equipped with a diamond-attenuated total reflection (ATR) crystal. Frozen saliva samples were thawed at room temperature (25 °C) for half an hour and the thawed saliva samples were homogenized using a vortex (2000 rpm for 20 s) prior to the MIR measurement. The ATR crystal was fully covered with the homogenized saliva sample (approx. 5 µL). Samples were immediately scanned, and spectra recorded. Each spectrum was computed using an average of 24 co-added interferograms at a resolution of 4 cm^−1^. A spectrum of air was used as a background prior to sample measurement and the spectrum of water was also measured every 20 samples. The instrument was operated using the OPUS software (version 8.5.29, Bruker Optics, Ettilgen, Germany). After each measurement, the surface of the ATR crystal was cleaned utilizing a 70% *w*/*w* ethanol/water solution and wiped with tissue paper between samples. A total of 156 saliva samples were collected for further analysis.

### 2.4. Chemometric Analysis

Before chemometric analysis, the MIR data of the unstimulated saliva samples were pre-processed using a baseline correction and Savitzky–Golay second derivative (second polynomial order and 21 smoothing points) (The Unscrambler X, Camo, Oslo, Norway) [37,38]. The fingerprint region (1800 to 450 cm^−1^) was utilized to establish partial least squares regression (PLS) models to predict satiety and satiation in the saliva samples using the MIR spectra. Classification models for low and high satiety were also developed using PLS discriminant analysis (DA) where samples belonging to the low satiety group were identified with the number 1 and samples from the high satiety group were identified with the number 2. The threshold governed the choice to turn a projected probability or score into a class label. In this study, the threshold was set to 1.5. The PLS and PLS-DA models developed were validated using cross validation (leave-one-out) [39,40].

## 3. Results and Discussion

Figure 1A shows the average MIR spectrum of the collected saliva samples in the fingerprint region (1800 to 600 cm^−1^) and compares the low and high satiety groups, as well as the saliva collected from all participants where all the food types analyzed were included. The effect of food types on the high or low satiety responses were observed in the MIR spectra of the saliva collected. In both avocado and banana, the absorbances for the low satiety perceiver samples were lower than the high satiety perceivers in the fingerprint region (1800–600 cm^−1^). The low absorbances might have indicated that the low satiety perceivers generally tended to produce more diluted saliva than the participants in the higher perceivers group. The main differences in the absorption values were observed between 1336–1364 cm^−1^ and were associated with the stretching vibrations of the carboxyl groups COO and asymmetric C-N stretching, which corresponded with the amide III group [5,41,42,43,44]. At 1270 cm^−1^ this band could be associated with CO groups corresponding with the presence of esters, and around 1076 cm^−1^ was associated with the presence of glycosylated proteins and phosphorus-containing compounds [5,41,42,44]. Peaks around 1437 cm^−1^ and 1473 cm^−1^ were associated with vibrations of δ(CH_2_) groups corresponding with proteins, lipids, fatty acids, and polysaccharides. These peaks have also been reported as biochemical indicators for triene conjugates and superoxide dismutase, which are present in saliva [32]. The peak at 1542 cm^−1^ has been reported to be associated with amide II (δNH, νCN) groups corresponding with salivary seromucoids [32]. The peak at 1647 cm^−1^ could be associated with amide I corresponding with albumin in the saliva, whereas the peak at 1653 cm^−1^ has been reported to be associated with amide I proteins in an α-helix conformation for salivary proteins [32]. The peak at 1717 cm^−1^ was associated with amide I purine bases, DNA, and RNA [5,32,41,42,44]. 

It has been reported [32] that the ratios between specific absorption bands in the spectra of saliva are correlated with salivary biochemical indicators (e.g., total protein, α-amino acids, lactate dehydrogenase). Figure 1B–D shows the ratios between frequencies at specific peaks when comparing high and low satiety perceivers for each of the three food types analyzed. When sorting out the saliva-satiety data according to the specific food types, differences could be identified. The low satiety perceivers had higher values for ratios 1542/2060 cm^−1^, 1637/3097 cm^−1^, and 1637/1616 cm^−1^ for avocado (*p* < 0.05) than the high satiety perceivers. Although differences in the same direction were found for apple and banana, these were not statistically significantly different. Similar results were reported by other authors where the ratio between 1542/2060 cm^−1^ was associated with total proteins based on the amide II group band and SCN^−^ thiocyanate in the saliva, between 1637/3097 cm^−1^ and 1637/1616 cm^−1^, was associated with α-amino acids and chlorides, respectively [32]. Although saliva from low satiety perceivers was more diluted (more water), it had an apparently higher percentage of protein and amino acids compared with other salivary organic components. The observed differences in the spectra of the saliva might have indicated that compositional variations in human saliva may be a result of underlying factors related to satiety perception types. One explanation could be that the saliva of low satiety individuals was more watery or diluted, but the concentration of proteins, α-amino acids, or chlorides, was higher compared to the high satiety saliva samples [33]. 

Table 1 shows the PLS cross validation calibration statistics for the prediction of satiety and satiation using unstimulated saliva collected from healthy participants. The coefficient of determination in cross validation (R^2^) and standard error in cross validation (SECV) reported for the prediction of satiety in all samples was 0.62 and 225.7 AUC, respectively. In contrast, poor calibration models were obtained for the prediction of satiation (R^2^ < 0.20) in all samples. These results might have reflected the nature of the experiment in which participants were asked to eat until comfortably full; although, there were differences in satiation responses between individuals where the term ‘comfortably’ full was interpreted differently by individuals. The lack of correlation with the MIR spectra suggested that the different responses were unlikely to be due to differences in oral physiology, as was reflected in the saliva composition. Calibration models were also developed for the different food types used. The R^2^ and SECV obtained for the prediction of satiety after consuming a banana was 0.63 and 188.1 AUC, respectively. However, poor PLS calibration models were obtained for the prediction of satiety using saliva samples collected from either the apple or avocado experiments (R^2^ < 0.20). 

The highest PLS loadings (Figure 2) for the prediction of satiety using all samples were observed around 1750 cm^−1^, which was associated with phospholipid, lipid, and ester groups [45,46], between 1665–1616 cm^−1^ was associated with amide I groups (proteins), 1286 cm^−1^ was associated with amide III groups, 1247 cm^−1^ was associated with PO_2_ of phosphate, 1089 cm^−1^ was associated with the symmetric stretching of phosphate groups of phosphodiester, 989 cm^−1^ was associated with C-O of ribose and C-C bonds, and 957 cm^−1^ was associated with polysaccharides [5,41,42,43,44]. Ni et al. [16] reported that the MIR frequencies between 1766–1725 cm^−1^ and 1692–1632 cm^−1^ were the most important when MIR calibrations were developed for the prediction of saliva flow, oral processing time, and fungiform papillae density of tongue. Other authors have reported that these oral physiology variables contributed to explaining satiety [15,22,47]. 

Previous studies have also reported on the prevalence of the low satiety phenotype groups in humans [30,31]. These researchers indicated that the presence of this phenotype group could be associated with stress and anxiety or lower blood cortisol responses to the meal. As described in Section 2, in this study, two groups were defined based on the satiety values (low and high). The R^2^ and SECV obtained for the prediction of the low and high satiety group was 0.92 and 0.10, respectively. The PLS-DA results showed that 100% and 98% of the saliva samples were correctly classified as low and high satiety perceivers, respectively. 

Overall, this study showed that the use of MIR spectroscopy provided a practical tool to understand the complex relationships between human physiology and self-reported responses based on human-food interactions. The results also showed that a relationship between saliva composition and satiety existed, although the quantitative models for the prediction of satiety were not robust. However, the use of PLS-DA models allowed reliable identification of saliva samples sourced from participants having low or high satiety responses. These models also indicated that MIR spectroscopy could be used for pre-selection or screening of participants in appetite sensory studies, reducing the time and cost of these types of studies. 

Some of the underlying factors that might influence the performance of the calibration models could be attributed to the fact that saliva itself was only one of many potential factors that could be considered to evaluate the human status to predict appetite. The human experience of appetite is not only decided by the human condition but is also influenced by the environment and whether the experiment design and saliva collection protocols were adequate to evaluate human–food interactions using MIR spectroscopy. Another important factor to consider was the utilization of unstimulated saliva. In this study, unstimulated saliva was used; however, it was possible that the use of stimulated saliva (e.g., after exposure to a specific chemical or mechanical stimulus) would result in alternate, or possibly more targeted, information on food-satiety interactions using MIR spectroscopy. 

## 4. Conclusions

Results from this study demonstrated the ability of MIR spectroscopy combined with chemometrics (e.g., PLS) to predict satiety from resting (unstimulated) saliva samples. Although quantitative PLS calibration models were not achieved, a qualitative model for the prediction of low and high satiety perception type was obtained using PLS-DA. Furthermore, this study indicated the possibility of evaluating the interactions between saliva and food using MIR spectroscopy as a rapid and cost-effective tool.

## Figures and Tables

**Figure 1 foods-11-00711-f001:**
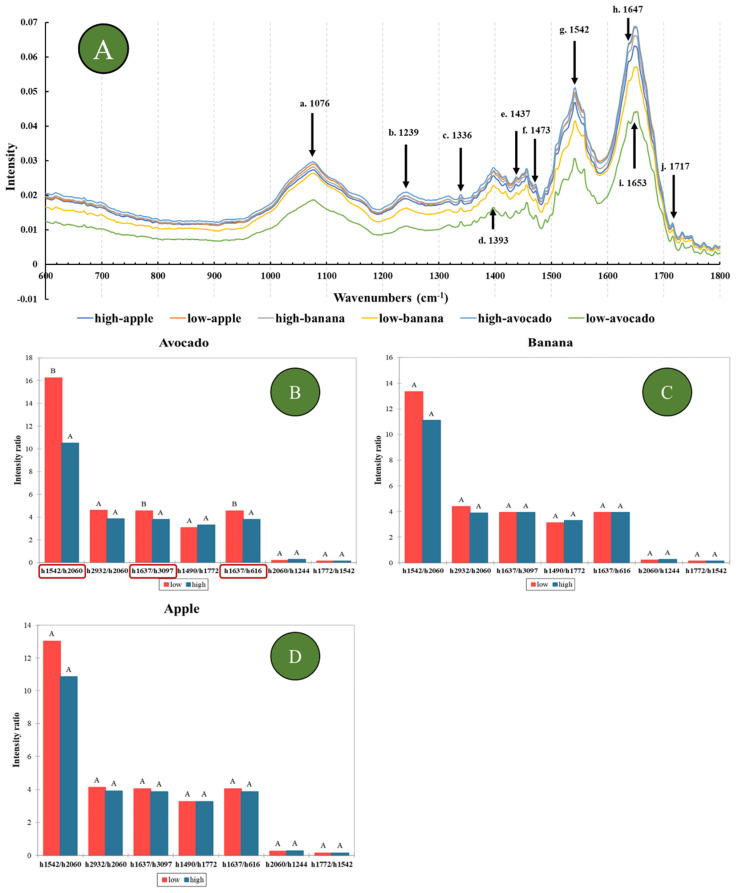
Mid-infrared spectra and ratios at specific frequencies of the unstimulated saliva samples analysed to show differences between low and high satiety groups. (**A**) Fingerprint region of salivary spectra comparing both food types and satiety perception types. The main reported absorption peaks in the literature [5,32,33,41,42,43,44,45,46] were labelled with lower case and wavelength number (a1076 cm^−1^, glycosylated proteins and phosphorus-containing components; b1239 cm^−1^, amide III/phospholipids; c1336 cm^−1^, carboxyl groups COO and asymmetric C-N stretching; d1393 cm^−1^, asymmetric and symmetric CH2 bending; e1437 cm^−1^, and f1473 cm^−1^, δ(CH_2_) groups corresponding to biochemical indicators for triene conjugates and superoxide dismutase; g1542 cm^−1^, amide II (δNH, νCN) groups; h1647 cm^−1^, amide I corresponding with albumin; i1653 cm^−1^, amide I proteins in α-helix; and j1717 cm^−1^, amide I purine bases, DNA and RNA). (**B**) Avocado, (**C**) banana, and (**D**) apple; ratios at specific frequencies calculated from the salivary spectra comparing the high and low satiety perceiver groups. The capital letters (e.g., A and B) in the figures signify significant difference between satiety perception groups.

**Figure 2 foods-11-00711-f002:**
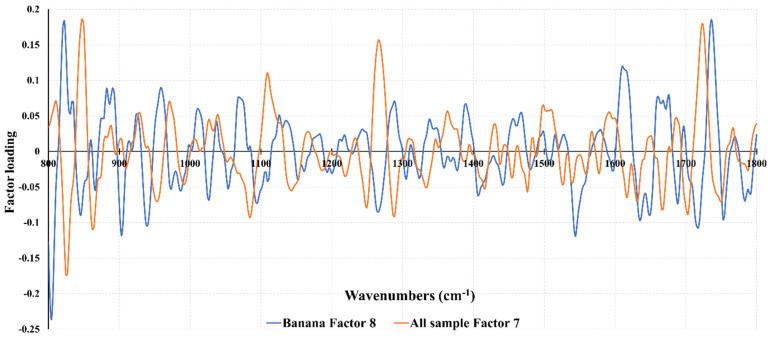
Partial least squares loadings derived from the calibration models used to predict satiety in the saliva of all samples or in the banana samples.

**Table 1 foods-11-00711-t001:** Descriptive statistics, partial least squares regression cross validation statistics for the prediction of satiety in saliva samples, and the PLS-DA cross validation statistics for the classification of saliva as low or high satiety.

	All Foods	Banana	Avocado	PLS-DA
R^2^	0.62	0.63	0.20	0.92
SECV	225.7	188.1	237.5	0.10
Bias	4.72	−12.5	0.60	0.001
Slope	0.67	0.62	0.20	0.97
LV	7	8	1	11
Mean (AUC)	1363	1456	1368	
SD	409	472	319	
Range	3138–423	3138–707	2272–525	

PLS-DA: partial least squares discriminant analysis; R^2^: coefficient of determination in calibration (R^2^); SECV: standard error in cross validation; SD: standard deviation; LV: number of latent variables used to develop the models.

## Data Availability

Not applicable.

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
