# Peer review of "Predicting Satiety from the Analysis of Human Saliva Using Mid-Infrared Spectroscopy Combined with Chemometrics"

_foods, 2022, doi:10.3390/foods11050711_

Round 1

Reviewer 1 Report

foods-1612464   TITLE: Predicting satiety from the analysis of human saliva using mid infrared spectroscopy combined with machine learning tools.   The manuscript is interesting and, in my opinion, it should be made minor modifications.    1) TITLE: machine learning refers to specific statitics. The use of this term in the title is quite misleading to the reader. Please remove it.   2) Line 12: Abstract: Machine learning (same as for the title).  3) Line 17: Abstract:  meaning of AUC should be given. 4) Line 104: at instead of ate 5) Line 108: Authors shoud explain what "Unstimulated" means. 6) Line 127: Authors should explain why three plant-based foods were used instead of three different categories of foods  7) Line 139: published (typing error)      

Author Response

Manuscript ID: foods-1612464

Predicting satiety from the analysis of human saliva using mid infrared spectroscopy

Dongdong Ni, Heather E Smyth, Michael J Gidley, Daniel Cozzolino*

We would like to thank the reviewers for their informative comments and the time they have taken to review this manuscript. Below we have addressed each of the comments and requests put forward by the reviewers, with a view to improving the manuscript.

To facilitate the reading, our comments and responses are in  italics. Changes made to the manuscript text are shown in tracked changes.

Reviewer 1

The manuscript is interesting, and, in my opinion, it should be made minor modifications.   

1) TITLE: machine learning refers to specific statistics. The use of this term in the title is quite misleading to the reader. Please remove it.  

We have removed the machine learning.

2) Line 12: Abstract: Machine learning (same as for the title). 

We have removed the machine learning.

3) Line 17: Abstract:  meaning of AUC should be given.

The AUC means the satiety area under the curve (AUC). We have added the definition before the abbreviation in the manuscript.

4) Line 104: at instead of ate

We have corrected the typing.

5) Line 108: Authors should explain what "Unstimulated" means.

We have added the explanation for the unstimulated saliva in the manuscript. “Unstimulated saliva is the saliva that continuously bathing oral cavity without chewing and taste stimulation.”

6) Line 127: Authors should explain why three plant-based foods were used instead of three different categories of foods.  

The reasons for using three plant-based foods in this study are following. Firstly, we aim to use the plant-based foods in this experiment. Because plant-based food (e.g. vegetables, fruits) takes up over two thirds of the everyday diet recommendation worldwide. Many benefits from keeping a diet rich in plant-based food are known such as reducing the risk of obesity and chronic diseases. The key features of the plant-based food include high dietary fibre, and cell wall structuring, induce strong satiation and long-time lasting satiety. It makes them the ideal choices for maintaining a healthy weight.  Secondly, we select three types of plant-based foods (apple, banana, and avocado). These three foods are selected as they are different in the source of energy provided and three main nutrients covered such soluble sugars for apple, starch for banana, and lipids for avocado. The different sources of energy and nutrients induce the variance on the satiation and satiety. This design not only balances the variations from the variety of main nutrients and energy source when we analyse three foods together, but also compares the differences in food and nutrient types when we analyse in each food type.  We have added more explanation in the paper’s methods part (2.2. satiation and satiety measurements).

7) Line 139: published (typing error).    

We have corrected the typing.

Reviewer 2 Report

Authors have combined IR spectroscopy with machine learning to establish a standard method for satiation and satiety in humans. For that, the authors have taken 52 healthy participants for the sample collection and PLS models were used to predict satiety and satiation from the collected saliva samples analyzed using the MIR spectra. Though the study is of the reader’s interest; however, I have several serious concerns that should be considered.

  1. Do authors have also considered some inclusion parameters for the selection of volunteers?
  2. Authors have considered individuals aged between 18 to 70 years. In my opinion, the groups can be divided into classes of age and the data can be analyzed accordingly.
  3. Authors have low and high satiety perceiver groups, does age have any effect on it?  If yes, please
  4. How many runs were considered on each sample in the Mid-infrared spectra collection for saliva to avoid the instrument error?
  5. Authors have considered the finger-print region (1800 to 450 cm -1) to establish partial least squares regression (PLS) models to predict satiety and satiation in the saliva samples using the MIR spectra. What was the reference to establishing the corresponding peaks to the respective?
  6. There are several peaks around 1000-1400 nm, please defined them.
  7. The calculated PLS, i.e., 0.62, 0.63, and 0.20 are too low for the accuracy of the model. I would recommend authors improve the data curation to increase the respective PLS values to the most acceptable values.
  8. Please explain the legend in detail and provide labels on the y-and x-axis. Figures should be rendered at higher resolution as well as labels or text mentioned in the figures should be increased for better legibility. 

Author Response

Manuscript ID: foods-1612464

Predicting satiety from the analysis of human saliva using mid infrared spectroscopy

Dongdong Ni, Heather E Smyth, Michael J Gidley, Daniel Cozzolino*

We would like to thank the reviewers for their informative comments and the time they have taken to review this manuscript. Below we have addressed each of the comments and requests put forward by the reviewers, with a view to improving the manuscript.

To facilitate the reading, our comments and responses are in italics. Changes made to the manuscript text are shown in tracked changes.

Reviewer 2

Authors have combined IR spectroscopy with machine learning to establish a standard method for satiation and satiety in humans. For that, the authors have taken 52 healthy participants for the sample collection and PLS models were used to predict satiety and satiation from the collected saliva samples analysed using the MIR spectra. Though the study is of the reader’s interest; however, I have several serious concerns that should be considered.

  • Do authors have also considered some inclusion parameters for the selection of volunteers?

The selection criteria for participants are clearly described in the methods part (2.1 participants and saliva collection). 52 healthy participants were recruited through open advertisement around Brisbane city. The selection criteria for participants were based on the following parameters, aged between 18 to 70 years, lack of oral cavities or dental diseases, no diabetes, not being pregnant or lactating at the time of the experiments, and not being diagnosed with psychological diseases like depression.

  • Authors have considered individuals aged between 18 to 70 years. In my opinion, the groups can be divided into classes of age and the data can be analysed accordingly.

We appreciated reviewer’s suggestion, but the effects of age to the satiety is another story which is not the key point of this paper. The aim of this study is to explore the relationship between saliva and satiety using mid infrared spectroscopy. We found the salivary differences between the low and high satiety perceivers. Based on the salivary differences we established an excellent qualitative PLS model to identify the low or high satiety perception types.

  • Authors have low and high satiety perceiver groups, does age have any effect on it?  If yes, please

Other researchers and published papers has reported that age might influence satiety. Additionally, some researchers reported that the low satiety perception phenotype has strong effects on the satiety perceptions.  However, few papers reported the confounding effects between the age and the satiety perception types. We think it is interesting and worth doing to design experiments (balanced both the age and satiety phenotype participants) to explore this point in the further studies, however it is not the aim of this study.  Moreover, we also double checked the age distribution in both low and high satiety perceiver groups during the data analysis. We found both the young and old aged participants showed the low and high satiety perception. So, we focused on the relationships between saliva and satiety instead of the confounding effects of age and satiety phenotype.

  • How many runs were considered on each sample in the Mid-infrared spectra collection for saliva to avoid the instrument error?

We collect have collected the saliva in triplicate, for each participant as biological repeats and we measure each sample once in the mid infrared spectroscopy. The result showed the error between the biological repeats is low. Moreover, the spectra of water were acquired between every 20 samples to guarantee the good stability of the equipment. In addition, prior to experiments, we have compared the error for the mid infrared instrument repeats. The results of the instrument are very stable and extremely low in error.

  • Authors have considered the finger-print region (1800 to 450 cm -1) to establish partial least squares regression (PLS) models to predict satiety and satiation in the saliva samples using the MIR spectra. What was the reference to establishing the corresponding peaks to the respective?

It is well known that the fingerprint region of saliva is around 1800 to 450 cm -1 and this study used this region to interpret the relationship between saliva and satiety. A series of references were cited in this paper and the main related ones (the corresponding peaks to the respective) are listed below.

  1. Rodrigues, R.P.; Aguiar, E.M.; Cardoso-Sousa, L.; Caixeta, D.C.; Guedes, C.C.; Siqueira, W.L.; Maia, Y.C.P.; Cardoso, S.V.; Sabino-Silva, R. Differential Molecular Signature of Human Saliva Using ATR-FTIR Spectroscopy for Chronic Kidney Disease Diagnosis. Braz Dent J 2019, 30, 437-445, doi:10.1590/0103-6440201902228.
  2. Rodrigues, L.M.; Magrini, T.D.; Lima, C.F.; Scholz, J.; Martinho, H.D.; Almeida, J.D. Effect of smoking cessation in saliva compounds by FTIR spectroscopy. Spectrochim Acta A 2017, 174, 124-129, doi:10.1016/j.saa.2016.11.009.
  3. Naseer, K.; Ali, S.; Qazi, J. ATR-FTIR spectroscopy as the future of diagnostics: a systematic review of the approach using bio-fluids. Appl Spectrosc Rev 2020, 56, 85-97, doi:10.1080/05704928.2020.1738453.
  4. Bel'skaya, L.V.; Sarf, E.A. Biochemical composition and characteristics of salivary FTIR spectra: correlation analysis. J Mol Liq 2021, 117380, doi:10.1016/j.molliq.2021.117380.
  5. Bel'skaya, L.V.; Sarf, E.A.; Solomatin, D.V. Age and Gender Characteristics of the Infrared Spectra of Normal Human Saliva. Appl Spectrosc 2020, 74, 536-543, doi:10.1177/0003702819885958.
  6. Ni, D., Smyth, H. E., Gidley, M. J., & Cozzolino, D. Towards personalised saliva spectral fingerprints: Comparison of mid infrared spectra of dried and whole saliva samples. Spectrochimica Acta Part A: Molecular and Biomolecular Spectroscopy 2021, 253, 119569. doi:10.1016/j.saa.2021.119569.

  • There are several peaks around 1000-1400 nm, please defined them.

We have added more labels for the peaks around 1000-1400 nm.

(a1076 cm-1, b1239 cm-1, c1336 cm-1, d1393 cm-1, e1437 cm-1,  f1473 cm-1, g1542 cm-1, h1647 cm-1, i1653 cm-1, and j1717 cm-1)

  • The calculated PLS, i.e., 0.62, 0.63, and 0.20 are too low for the accuracy of the model. I would recommend authors improve the data curation to increase the respective PLS values to the most acceptable values.

We appreciated the reviewer’s suggestion. However, the PLS models in this paper are only used to interpret the relationship between saliva and perceived satiety, not for the quantitative prediction. The value of coefficient of determination (R2, 0.62, 0.63) represents the intensity of connection between saliva and perceived satiety. It indicates there are connections between saliva and satiety, but these connections are not strong enough to establish an accurate quantitative prediction model, which is the reason we developed an excellent qualitative model for the prediction of low and high satiety perception types using PLS-DA (R2, 0.92).

In addition, we used self-reported perceived satiety as independent variable to develop the quantitative PLS regression model. However, the deviation for the self-reported satiety method is very large. It may be another important reason for the low accuracy of the PLS models.  Further, more studies are needed to establish a robust model for satiety using saliva with a larger number of participants and food types. Also, more accurate methods are needed to measure the satiety.

  • Please explain the legend in detail and provide labels on the y-and x-axis. Figures should be rendered at higher resolution as well as labels or text mentioned in the figures should be increased for better legibility. 

We have improved the figures and added the labels.